# Clinical Significance of the Plasma Biomarker Panels in Amyloid-Negative and Tau PET-Positive Amnestic Patients: Comparisons with Alzheimer’s Disease and Unimpaired Cognitive Controls

**DOI:** 10.3390/ijms25115607

**Published:** 2024-05-21

**Authors:** Hsin-I Chang, Kuo-Lun Huang, Chung-Gue Huang, Chi-Wei Huang, Shu-Hua Huang, Kun-Ju Lin, Chiung-Chih Chang

**Affiliations:** 1Department of Neurology, Cognition and Aging Center, Kaohsiung Chang Gung Memorial Hospital, Chang Gung University College of Medicine, Kaohsiung 83301, Taiwan; homelover@gmail.com (H.-I.C.); justin1124@cgmh.org.tw (C.-W.H.); 2Institute for Translational Research in Biomedicine, Kaohsiung Chang Gung Memorial Hospital, Kaohsiung 83301, Taiwan; 3Department of Neurology, Linkou Chang Gung Memorial Hospital, Chang Gung University, Taoyuan 333423, Taiwan; drkuolun@gmail.com; 4Department of Medical Laboratory, Linkou Chang Gung Memorial Hospital, Department of Medical Bio-Technology and Laboratory Science, Chang Gung University, Taoyuan 333423, Taiwan; joyce@cgmh.org.tw; 5Department of Nuclear Medicine, Kaohsiung Chang Gung Memorial Hospital, Chang Gung University College of Medicine, Kaohsiung 83301, Taiwan; sophia4790@cgmh.org.tw; 6Department of Nuclear Medicine, Linkou Chang Gung Memorial Hospital, Chang Gung University, Taoyuan 333423, Taiwan; kunjulin@gmail.com; 7School of Medicine, College of Medicine, National Sun Yat-sen University, Kaohsiung 80404, Taiwan

**Keywords:** pTau181, tau first cognitive tauopathy, primary age-related tauopathy, florzolotau

## Abstract

The purpose of this study was to investigate whether plasma biomarkers can help to diagnose, differentiate from Alzheimer disease (AD), and stage cognitive performance in patients with positron emission tomography (PET)-confirmed primary age-related tauopathy, termed tau-first cognitive proteinopathy (TCP) in this study. In this multi-center study, we enrolled 285 subjects with young-onset AD (YOAD; *n* = 55), late-onset AD (LOAD; *n* = 96), TCP (*n* = 44), and cognitively unimpaired controls (CTL; *n* = 90) and analyzed plasma Aβ42/Aβ40, pTau181, neurofilament light (NFL), and total-tau using single-molecule assays. Amyloid and tau centiloids reflected pathological burden, and hippocampal volume reflected structural integrity. Receiver operating characteristic curves and areas under the curves (AUCs) were used to determine the diagnostic accuracy of plasma biomarkers compared to hippocampal volume and amyloid and tau centiloids. The Mini-Mental State Examination score (MMSE) served as the major cognitive outcome. Logistic stepwise regression was used to assess the overall diagnostic accuracy, combining fluid and structural biomarkers and a stepwise linear regression model for the significant variables for MMSE. For TCP, tau centiloid reached the highest AUC for diagnosis (0.79), while pTau181 could differentiate TCP from YOAD (accuracy 0.775) and LOAD (accuracy 0.806). NFL reflected the clinical dementia rating in TCP, while pTau181 (rho = 0.3487, *p* = 0.03) and Aβ42/Aβ40 (rho = −0.36, *p* = 0.02) were significantly correlated with tau centiloid. Hippocampal volume (unstandardized β = 4.99, *p* = 0.01) outperformed all of the fluid biomarkers in predicting MMSE scores in the TCP group. Our results support the superiority of tau PET to diagnose TCP, pTau181 to differentiate TCP from YOAD or LOAD, and NFL for functional staging.

## 1. Introduction

With successful treatments that promise to slow Alzheimer’s disease (AD), increasing research has focused on methods to improve the diagnosis of AD. According to the National Institutes on Aging and Alzheimer’s Association’s (NIA-AA) working framework for AD [1], amyloid and tau are involved in the upstream pathogenesis. In plasma biomarker spectra, amyloid beta (Aβ) 42, Aβ40, and phosphorylated tau181 (pTau181) are considered the core AD fluid biomarkers. Biologically, the Aβ42/Aβ40 ratio reflects Aβ plaque pathophysiology [2], and pTau181 is an indicator of ongoing tau pathology [3]. The neurofilament light (NFL) is considered a strong indicator of neurodegeneration [4,5,6]. Various blood assays have been developed for the detection of these plasma biomarkers [7,8,9,10]. In a clinical setting, the use of plasma biomarkers can be considered a more cost-effective way, and the validity of the biomarkers can increase the specificity before the arrangement of the confirmatory tool (i.e., amyloid or tau positron emission tomography [PET]).

The importance of tau pathology independent of Aβ has received increasing attention but remained largely unexplored. Tau in the absence of amyloid (A-T+) is considered to be a ‘non-AD pathology’ [11,12]. From the literature search, primary age-related tauopathy (PART) [13] is a pathological term showing an A-T+ biomarker status [14,15]. In one pathological report [16], plasma pTau181 was found to differentiate AD from PART; however, the evaluation of fluid biomarkers and cognitive staging in pathology series showed time lag. Other nomenclature, such as suspected non-AD pathophysiology [17] or the tau-first AD subtype [18], also shows A-T+ status. In amyloid-PET-negative individuals, CSF pTau181 levels were associated with concordant increases in tau-PET signals [19]. The use of CSF fluid biomarkers was of high specificity, and the increase of pTau181 in the CSF of amyloid-negative individuals supports its role on the tau pathway. Based on these reports, the use of pTau181 to differentiate AD versus A-T+ cases is possible but requires further validation on other plasma biomarkers.

Debate exists as to whether some patients with A-T+ status may have had tau deposition prior to amyloid, and several tau regulators participated in their neuronal injury cascades [18,20]. Others suggested that tau in the absence of amyloid may only be caused by age-related changes. Research methods relying on postmortem brain pathology, such as PART, may hinder the elucidation of tau pathology independent of Aβ pathology. The combination of amyloid- and tau-PET has allowed for the antemortem diagnosis of amyloid and tau status. Florzolotau (^18^F) is a second-generation tau tracer that can identify 3-repeat tau, 4-repeat tau, or a mixture of the two [21,22] and is probably the most comprehensive tool to access tau pathology in vivo. In this manuscript, we defined patients who had a clinical phenotype of AD that also fulfilled the in vivo PET-based A-T+ criteria as having tau-first cognitive proteinopathy (TCP). TCP is a novel term. The positive tau-PET neuroimaging characteristics and the negative amyloid status make TCP a great candidate to assess the role of tau independently of amyloid pathology.

The objective of this study was to explore whether predefined AD core plasma biomarkers (i.e., Aβ42/Aβ40 ratio, pTau181) and neuronal injury biomarkers (NFL, total tau) may provide clinical significance in the diagnosis, staging, and cognitive prediction of patients with TCP. In this study, we enrolled cognitively unimpaired controls (CTL) and patients with AD for comparisons. There were overlapping features between TCP and AD, such as amnestic features, volumetric atrophy in the medial temporal regions [23], and tau burden topography [24]. The discrimination between AD (A+T+) and TCP showing A-T+ status is critical from the perspective of amyloid-removing treatment. According to the biological properties of amyloid cascades, AD is different from TCP. We hypothesized that AD core plasma biomarkers may open a possible window for differential diagnosis purposes and that neuronal injury biomarkers may assist in clinical staging.

## 2. Results

### 2.1. Cohort Demographics

A total of 285 participants were enrolled (CTL 90, LOAD 96, TCP 44, and YOAD 55; Table 1). More of the patients with LOAD and YOAD were *ApoE4* carriers compared to those with TCP. More of the patients with TCP were *ApoE4* carriers and had a higher tau centiloid score compared to the CTL. The ages of the TCP and LOAD groups were not significant. Among the four diagnostic groups, there were no significant differences in plasma biomarker levels between the *ApoE4* carriers and non-carriers.

### 2.2. Age Effects on Plasma Biomarkers

The ranges of biomarker assays in the CTL, AD, and TCP are shown in Appendix A. Significant age effects on pTau181, Aβ42/40 ratio, and NFL (Appendix A) were observed in the CTL group. For AD, only NFL showed a positive age effect (Appendix A). For TCP, pTau181 and NFL showed a positive age effect (Appendix A).

### 2.3. Tau-PET Distribution

The topographical display of tau PET is shown in Figure 1A. The TCP group had more restricted tau deposition, and they were more prominent in the posterior brain regions. Comparisons between the LOAD and YOAD groups were not significant in tau-PET topography. An amyloid centiloid score of 25.5 showed the highest AUC (0.936) in differentiating AD from TCP (specificity = 0.911, sensitivity = 0.866). For tau centiloid, a level of 28 showed the highest AUC (0.976) in differentiating AD from TCP (sensitivity = 0.955, specificity = 0.951).

### 2.4. YOAD Was Associated with Faster Neurodegeneration Than LOAD

A total of 227 cases had at least three T1 images (CTL 65, LOAD 70, TCP 42, and YOAD 50). Based on the time effect of cortical thickness, the patients with YOAD showed more rapid neurodegeneration (Appendix A) than those with LOAD. We did not detect significant cortical thickness changes in the patients with TCP or in the CTL groups.

### 2.5. Diagnostic Value of Plasma pTau181 in TCP and AD

Plasma pTau181 levels were lower in TCP, and the levels were significantly lower than in the YOAD or LOAD groups (Figure 1B). pTau181 in AD had a higher fold change compared with TCP (Appendix A, fold change = 1.89). The TCP group had a higher NFL level (Appendix A, fold change = 1.51) than the CTL group, and the NFL levels in TCP and AD were not statistically different.

Levels of pTau181, Aβ42/40, and NFL significantly differentiated the two AD groups from the CTL group (Figure 1B), among which pTau181 had the highest fold changes in AD (Appendix A, fold change = 2.16). The total tau level was significantly higher in the LOAD group than the CTL group but was comparable between the YOAD and CTL groups.

We transformed the fluid biomarker levels into Z scores to better explore the pattern differences in AD and TCP (Figure 1C). The results showed that the YOAD and LOAD groups shared similar biomarker patterns of low Aβ42/40 and high pTau181 profiles. All three disease groups had high NFL levels.

### 2.6. ROC Curves of Image- or Plasma-Biomarker

Using the CTL as the reference group, we plotted ROC curves for the three disease groups (Figure 2). For two AD groups, tau centiloid had the highest AUC values for diagnosis, followed by amyloid centiloid, hippocampal volume, pTau181, Aβ42/40 ratio, and NFL (*p* < 0.05). For the TCP group, tau centiloid, hippocampal volume, and the Aβ42/40 ratio reached statistical significance. The combined model using tau centiloid, hippocampal volume, and the Aβ42/40 ratio additionally increases the AUC to 0.82.

### 2.7. Stepwise Logistic Regression Model Using Hippocampal Volume and Four Plasma Biomarkers for Diagnosis or Differential Diagnosis

As the diagnosis of AD and TCP was based on the amyloid and tau image readouts, we further assessed whether the combined use of the hippocampal volume or plasma biomarkers may provide better diagnostic accuracy than the pathological images (Table 2). For TCP/CTL, only hippocampal volume reached statistical significance. The composite of hippocampal volume and plasma pTau181 level assisted in diagnosing LOAD/CTL with an accuracy of 0.830, while Aβ42/40 values in the YOAD group additionally increased the diagnostic accuracy to 0.846.

The differential diagnosis model is shown in Table 2. Plasma pTau 181 level showed statistical significance to differentiate the TCP group from the LOAD or YOAD, while the hippocampal volume increased the diagnostic accuracy to 0.806 in the LOAD/TCP.

### 2.8. Associations of Plasma Biomarkers with Amyloid and Tau PET Centiloid

After adjusting for age in the TCP group, Aβ42/40 (rho = −0.36, *p* = 0.02) and pTau181 (rho = 0.35, *p* = 0.03) values both correlated with tau centiloid. For AD, pTau181 (rho = 0.3, *p* = 0.001) and NFL (rho = 0.223, *p* = 0.02) levels were related to tau centiloid, and pTau181 (rho = 0.2295, *p* = 0.01) level was related to amyloid centiloid.

### 2.9. Plasma Biomarkers for Clinical Staging

In the TCP group, NFL level was significantly related to the increment in CDR score. For AD, both NFL and Aβ42/40 values were related to the increment of the CDR score (Figure 3).

### 2.10. Cognitive Prediction Model

We further assessed the predictive role of biomarkers on MMSE scores in each group using a stepwise linear regression model (Table 3). In the TCP group, the only predictor was hippocampal volume. In the AD group, hippocampal volume, tau centiloid, and educational years were significant. The predictors for LOAD and YOAD were different. The correlations among plasma biomarker levels with MMSE and CASI scores are shown in Appendix A.

## 3. Discussion

We tested the clinical feasibility of using fluid biomarkers to diagnose, stage, and cognitively predict TCP. The results can be summarized as follows: First, pTau181 levels could help differentiate TCP from YOAD (accuracy = 0.78, Table 2), and the combination of pTau181 and hippocampal volume could differentiate TCP from LOAD with an accuracy of 0.81 (Table 2). Second, we found that pTau181 levels were significantly correlated with tau centiloid in the patients with TCP (result Section 2.8), supporting its role in monitoring tau burden. Third, we confirmed the clinical significance of using plasma biomarkers to reflect the clinical stage (Figure 3) and cognitive test scores (Table 3). The correlations between MMSE score with pTau181 (Appendix A) and NFL (Appendix A) were significant in the patients with TCP, while NFL levels were correlated with higher CDR scores (Figure 3). However, the effect of hippocampal volume still outweighed these plasma biomarkers in stepwise regression analysis (Table 3).

In this study, of all biomarkers, tau centiloid had the highest diagnostic accuracy for TCP (AUC = 0.789, Figure 2), followed by hippocampal volume (AUC = 0.716). Using the Florzolotau (18F) radiotracer, intraneuronal tau filament binding sites have been demonstrated in PART [25] and AD [22]. The use of tau-PET to diagnose TCP in amyloid-β negative cases has been shown to be possible using tau tracers such as ^18^F-RO948 [19], Florzolotau (18F) radiotracers, (18)F-Flortaucipir, and (18)F-MK-6240 [26]. As there are no clinical criteria for diagnosing TCP, the diagnostic workflow for TCP should be slowly built toward the use of in vivo PET [27].

Among the biomarkers panel, NFL is the only biomarker showing level differences between TCP and CTL (Figure 1B). Stratified by CDR scores, the NFL levels in the TCP group also increased in a dose-related manner, supporting its role in monitoring clinical stages (Figure 3). Although the tau burden is salient in TCP, the increase in NFL may reflect co-pathologies other than tau, as the correlations between tau centiloid and NFL were not significant. *ApoE4* carrier status has been reported to be low in patients with TCP [11,13,28,29,30,31]. Although *ApoE4* carrier status was lower in our patients with TCP than in those with AD, it was still higher than in the CTL. The dual-pathway theory suggests that *ApoE4*, cholesterol metabolism, the endocytic system, and microglial activation also mediate amyloid-independent cascades in TCP [20]. Coexisting pathologies such as TDP-43 have been reported in patients with TCP, and they may augment atrophy in the amygdala, hippocampal, and anterior temporal regions [32].

There is no significant difference in plasma Aβ42/40 levels between the TCP and AD groups (Figure 1B). As the Aβ42/40 ratio indicates dysregulated Aβ metabolism and processing, the lack of differences between TCP and AD in the Aβ42/40 ratio requires further exploration. Around 9% of patients with TCP who are initially amyloid-negative on PET may subsequently become amyloid-positive [33], suggesting the low detection ability of amyloid PET at subthreshold levels or the synergic effect of tau on amyloid beta accumulation [34]. Another possibility is an association between the Aβ42/40 ratio and tau pathology, as we found that the Aβ42/40 ratio was inversely correlated with tau centiloid levels in the TCP group. In a large cohort study pooling various neurodegenerative disorders, the CSF Aβ42/40 ratio was strongly associated with tau markers [35]. As TCP is defined as an amyloid-negative pathology, whether the Aβ42/40 ratio in TCP reflects subthreshold amyloid burden, tau, or other co-pathologies requires more evidence.

In our cognitive model, tau centiloid was not predictive of MMSE in the patients with TCP, and hippocampal volume was the only significant independent variable when the fluid biomarkers were entered into the statistical model (Table 3). Tau burden in TCP often extends beyond the hippocampus, as shown in our study and others [33,36]. The influence of tau on volumetric atrophy in TCP seems to target the medial temporal region [36], and the influence of hippocampal atrophy on cognition outweighs fluid biomarkers.

TCP is not uncommon in clinical settings [37], and the clinical features of TCP may mimic AD, manifesting as impaired short-term memory. Previously, tau in the absence of amyloid was considered to be age-related; however, the age at disease onset in our TCP group was not restricted to older patients (ranged 49~85 years, Table 1). Individuals classified as having PART are mostly cognitively unimpaired; however, our TCP patients had a CDR score ranging from 0 to 2. The patients in previous PART studies were diagnosed by pathological labeling of tau and amyloid, while we used imaging criteria. Therefore, the pathological definition may not cover the same study population as in our study using in vivo imaging criteria.

We found that plasma pTau181 levels were significantly lower in our TCP patients compared to the YOAD and LOAD groups (Figure 1B). Previous studies have shown that higher concentrations of plasma pTau181 are associated with amyloid pathology [38,39] and that changes in the concentration could be used to differentiate AD from TCP. CSF pTau181 has been associated with tau-PET measures in TCP [19], and we also found that plasma pTau181 levels were correlated with tau centiloid in TCP. The transition from CSF to blood-based pTau181 markers to reflect the pathological burden of tau may be feasible in TCP.

With the success of amyloid-removing therapy in AD, using plasma pTau181 for trial selection prior to PET imaging may be beneficial [40]. In clinical practice, identifying an appropriate age-related reference range is critical, as pTau181 levels showed an age effect in our CTL. The ability to differentiate TCP from AD still relies on higher pTau181 levels in AD. The Aβ42/40 ratio may also be able to differentiate TCP from AD, as it can be binarized by amyloid PET scans. However, the significance was not established in this study.

Although disease onset, progression, and genetic predisposition are the basic differences between YOAD and LOAD, they share similar pathological entities. In head-to-head comparisons of the YOAD and LOAD groups here, plasma biomarker profiles were similar. For both AD groups, pTau181 had the highest discrimination ability from the CTL (AUC = 0.82~0.86), followed by NFL and Aβ42/40 ratio (Figure 2). As these plasma biomarkers become abnormal ahead of PET image findings, these three biomarkers could be used to accurately identify AD pathology versus biomarker-negative CTL [41,42,43,44,45].

We found that tau centiloid had a higher AUC than amyloid to diagnose YOAD and LOAD (Figure 2). With advances in fluid biomarkers, tau PET may be sufficient to confirm the diagnosis and monitor the severity of the disease. The use of tau centiloid to differentiate AD from TCP may also be possible based on our results. The low tau centiloid level in TCP here was directly related to the region of interest in this study, as the topography of TCP was not overlapping with AD. From the patient resilience perspective, we propose that appropriate selection and cutoff values using a fluid biomarker panel followed by a confirmative amyloid or tau PET study may be suitable to diagnose patients with AD for amyloid-modifying treatment [46,47].

Our results also showed the robustness of the NFL to monitor clinical stages (Figure 3). Regarding the diagnosis, NFL was the only plasma biomarker that could differentiate YOAD from LOAD, with moderate accuracy (AUC = 0.662). We also found greater cortical thickness degeneration in the patients with YOAD, although the average NFL in the YOAD group was lower. As NFL showed an age effect, the increase in NFL in the patients with LOAD may have been driven by age.

This study has several limitations. First, the report on plasma biomarkers was based on a cross-sectional observational study based on a neuroimaging cohort. Therefore, the comparisons of fluid biomarkers and clinical stages and the relationships with pathological burden may not reflect disease progression patterns. The robustness of these AD core fluid biomarkers, however, was replicated in our study population, with patients enrolled from the north to the south of Taiwan. A longitudinal follow-up study will further our understanding of the applicability of fluid biomarkers to monitor disease progression. Second, as there is currently no consensus on the clinical criteria for PART, the diagnosis of TCP in this study may not be the same as reports using pathology. The amyloid-negative status in the patients with TCP may have been mixed with those showing focal or subthreshold amyloid deposition. As we also included patients with LOAD or YOAD for comparison, the appearance of tau burden prior to amyloid on in vivo PET still deserves clinical attention. The ability of pTau181 to discriminate AD from TCP also supports certain biological differences. Finally, various assays and combinations are available for fluid biomarkers and PET radiotracers to diagnose AD or in vivo PART. Direct comparisons between our study and others will be difficult. As such, we only used amyloid and tau centiloid to reflect the pathological burden. In addition, cutoff values for the fluid biomarkers were not defined as the sample size was small. The enrollment of more controls and understanding of confounding factors other than age and *ApoE4* status may help broaden the diagnostic repertoire.

## 4. Materials and Methods

### 4.1. Patient Enrollment

This study was conducted in accordance with the Declaration of Helsinki and was approved by the Institutional Review Board of Chang Gung Memorial Hospital. All individuals in this study were enrolled from the cognition and aging cohort [48,49]. The cognition and aging cohort is an ongoing clinical-neuroimaging study of participants with memory complaints that was initiated in 2006. All participants were enrolled at the Chang Gung Memorial Hospitals in Kaohsiung, Fen Shan, Linkou, and Taipei City, while cognitively unimpaired CTL were recruited from the community. The study workflow is shown in Appendix A.

### 4.2. Diagnosis of AD

Participants who fulfilled the updated NIA-AA 2018 criteria [1] were considered to have AD [50]. Episodic memory deficits were defined using the Chinese version of the Verbal Learning Test [51]. Amyloid- and tau-PET positivity were diagnosed according to visual readouts and the consensus of two independent nuclear medicine specialists. We further divided the AD patients into young-onset AD (YOAD; age at onset < 65 years) and late-onset AD (LOAD; age at onset ≥ 65 years) groups.

### 4.3. Diagnosis of TCP

Patients with TCP were diagnosed based on in vivo PET criteria with negative amyloid and positive tau readouts [19] and clinical presentations showing amnestic features. We excluded those who fulfilled the criteria of dementia due to frontotemporal dementia [52], diffuse Lewy body dementia, Parkinson’s disease dementia [53], progressive supranuclear palsy [54], multiple system atrophy [55], corticobasal syndrome [56], and small vessel disease.

### 4.4. Cognitively Unimpaired CTL

Neurologically and cognitively unimpaired CTL were enrolled according to the following inclusion criteria: (1) age ≥ 20 years; (2) not having cognitive symptoms as assessed by a behavioral neurologist using their judgment of symptoms on the National Alzheimer’s Coordinating Center B9 form; (3) not fulfilling the criteria for mild or major neurocognitive disorders according to the Diagnostic and Statistical Manual of Mental Disorders, fifth edition (DSM-V); and (4) a Mini-Mental State Examination (MMSE) score of 25–30 according to the educational level (≤9 years of education an MMSE score of 25–30; >9 years an MMSE score of ≥28).

The general exclusion criteria for the study were a history of clinical stroke during the screening phase, a modified Hachinski ischemic score of ≥4, clinically unmanaged diabetes, major depressive disorder, or dysthymic disorder.

### 4.5. Demographic and Cognitive Evaluations

The demographic data in this study included apolipoprotein E4 (*ApoE4*) status, estimated age at disease onset based on caregiver reports of first symptoms, years of education, and gender. Disease duration was calculated by subtracting the age at plasma sampling from the estimated age at disease onset. The obtained *ApoE4* genotypes were dichotomized into ε4 carriers (heterozygous or homozygous) and non-carriers (i.e., ε2 or ε3 carriers). The neurobehavioral tests included the MMSE and 9 subdomains of the Cognitive Abilities Screening Instrument (CASI) [57]. Total MMSE and CASI scores were considered to reflect a global assessment of cognitive function. We used the Clinical Dementia Rating (CDR) scale to reflect clinical staging. In this study, we selected imaging data with the closest acquisition date to the plasma collection (<3 months).

### 4.6. Blood Sample Collection

Fasting venous blood samples were obtained using EDTA as the anticoagulant at baseline. The supernatant plasma was aliquoted into 1 cc polypropylene tubes and frozen at −80 °C. All samples were transferred and stored at the Laboratory of Linkou Chang Gung Memorial Hospital.

### 4.7. Single-Molecule Array Analysis of Plasma Aβ42/40, pTau181, NFL, and T-Tau

Single-molecule array analysis was used for the ultrasensitive quantification of AD biomarkers (HD-X instrument, Quanterix, Billerica, MA, USA). Plasma Aβ1–42, Aβ1–40, and total tau levels were determined using a multiplex array (Neurology 3-Plex A Advantage Kit, N3PA, Quanterix, Billerica, MA, USA). NFL and pTau181 levels were measured (NFL single-analyte array, P-tau181 V2, Quanterix, Billerica, MA, USA). The lowest limit of quantification and pooled coefficients of variation for these assays were 2 pg/mL and 15.7% for pTau181, 0.0715 pg/mL and 18% for total-tau, 0.2 pg/mL and 18% for NFL, 0.225 pg/mL and 6% for Aβ1–42, and 1.9 pg/mL and 6% for Aβ1–40.

### 4.8. Image Acquisition and Processing

3D T1-weighted magnetization-prepared rapid gradient echo (MP-RAGE) was acquired on a 3T Siemens (Skyra, Erlangen, Germany) using the following parameters: 176 slices, repeat time: 2600 ms, echo time: 3.15 ms, inversion time: 1090 ms, flip angle: 13°, with 0.5 × 0.5 × 1 mm voxel size. Structural images were processed using the pipeline of FreeSurfer (version 7.1.1, https://surfer.nmr.mgh.harvard.edu/). The hippocampal volume was calculated as the average of the left and right hippocampuses.

### 4.9. Amyloid and Tau PET Acquisition

PET images were acquired on a GE Discovery MI PET/CT scanner (GE Healthcare, Chicago, IL, USA). Amyloid scans were acquired using ^18^F-florbetaben (at 90 min) or ^18^F-florbetapir (at 50 min). Florzolotau (F18) tau-PET scans were acquired at 90 min after the injection of 185 ± 74 MBq.

PET image acquisition consisted of two 5-min dynamic frames to allow for motion correction. Three-dimensional PET images were acquired, and the images were reconstructed with an iterative reconstruction algorithm (OSEM 4 iterations, 16 subsets). Low-dose CT scans for attenuation correction were acquired with the following parameters: 15 mAs, 120 keV, 512 × 512 matrix, 2.79-millimeter slice thickness, 71 slices, 110 mm/s increments, 0.5-s rotation time, and a pitch of 1.375.

For the amyloid images, the standard uptake value ratio (SUVr) reference was the whole cerebellum cortex. The amyloid centiloid score was calculated based on the Centiloid Project (https://www.gaain.org/centiloid-project accessed on 1 January 2023). For Florzolotau (18F), the SUVr was generated with the reference region in the white matter. Tau composite regions of interest included the anterior cingulate, frontal, precuneus, lateral parietal, and lateral temporal gyri.

The tau centiloid (CL) equation for an individual’s SUVr (SUVr_IND_) is as follows:$$\mathrm{CL}=\frac{{\mathrm{SUVr}}_{\mathrm{IND}}-0.912}{0.368}\times 100$$

### 4.10. Surface-Based Topography in Tau-PET

The preprocessing of surface-based analysis followed the workflow of PET-surfer (https://surfer.nmr.mgh.harvard.edu/). Spatially normalized Florzolotau (18F) SUVr maps were then convoluted using a Gaussian kernel with an 8-millimeter full width at half maximum to reduce noise. We used the following contrasts to display the tau topography: LOAD > CTL, YOAD > CTL, and TCP > CTL. With cluster-wise correction computed with parametric Gaussian-based simulations to compute a false positive rate of 0.001, we used the vertex-wise threshold of 3 [58].

### 4.11. Assessing the Neurodegenerative Process

For the three disease groups, we retrieved historical T1 images to calculate the time effects (months) of cortical thickness, and the results indicated the topography being affected by the neurodegenerative process. For longitudinal MRI analysis, we used the recon-long process in FreeSurfer (version 7.1.1, https://surfer.nmr.mgh.harvard.edu/) and the linear mixed-effect model [59] with MATLAB 2019b (The Mathworks Inc., Natick, MA, USA). The model considered cortical thickness as the dependent variable and age, gender, educational years, and estimated total intracranial volume (eTIV) as nuisance covariates. The threshold was a vertex-wise threshold of 3 [58].

### 4.12. Statistical Analysis

Statistical analysis was performed using R version 4.3.1 (https://www.r-project.org). Descriptive analyses were performed using the Kruskal-Wallis or Mann-Whitney U test with post-hoc comparisons, as appropriate. Categorical variables were analyzed using Fisher’s exact test. Spearman correlations were calculated between continuous variables and adjusted for confounders as needed. To address the biomarker panels for the LOAD, YOAD, and TCP groups, we calculated the Z score of each plasma biomarker using the mean and standard deviation of the CTL group.

To understand the diagnostic accuracy of the predefined biomarkers, we used receiver operating characteristic (ROC) curves for binary classifications (disease groups versus CTL). Tested biomarkers included tau centiloid, amyloid centiloid, hippocampal volume (average of the left and right hippocampus), pTau181, total tau, NFL, and Aβ42/40 ratio. Areas under the ROC curves (AUCs) of significant biomarkers were reported.

To understand the accuracy of differentiating AD (YOAD, LOAD) from TCP, a stepwise logistic regression model was used. As AD and TCP can be well defined by amyloid and tau PET, we only included four plasma biomarkers and the hippocampal volume in this model. The accuracy of the significant model and related parameters were reported.

To assess whether the plasma biomarkers could reflect pathological burden in the three disease groups, we used Spearman correlations with amyloid or tau centiloids, adjusted for age. The interval plot (mean, with 95% confidence interval) was used to explore each plasma biomarker level and the clinical stages.

Finally, we tested the best model for predicting cognitive test scores in the three disease groups using the stepwise linear regression model. For the cognitive tests, the MMSE score represented the dependent variable, and the independent variables were amyloid centiloid, tau centiloid, hippocampal volume, educational years, and four fluid biomarker values. A *p* value of <0.05 was considered to be statistically significant.

## 5. Conclusions

We used core AD biomarkers to assess their clinical feasibility in patients with TCP. Our results validated the clinical utility of plasma NFL to discriminate TCP from CTL, pTau181 to differentiate TCP from AD, and the use of hippocampal volume to predict cognitive scores. Both the Aβ42/40 ratio and pTau181 were closely correlated with tau centiloid in the patients with TCP, while NFL was correlated with CDR. Although the rate of progression may be different between patients with LOAD and YOAD, a combined panel of plasma pTau181, NFL, and the Aβ42/40 ratio could provide diagnostic specificity for screening purposes. The higher accuracy of tau centiloid than amyloid centiloid in diagnosing AD may support its use in the confirmative workflow once fluid biomarker screening processes have been established.

## Figures and Tables

**Figure 1 ijms-25-05607-f001:**
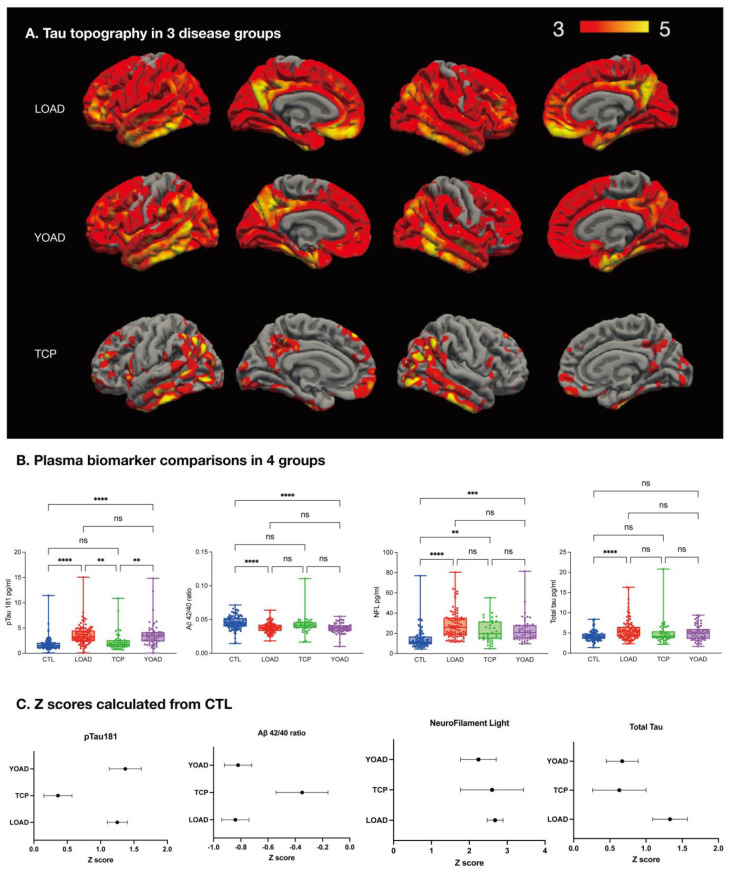
(**A**) Topographic display of Florzolotau (F18) in the three disease groups compared with the CTL; significance set as a vertex threshold of 3 with parametric Gaussian-based simulations and cluster-wise corrections. (**B**) Plasma biomarker comparisons in the four groups. (**C**) Z transformation showing disease-related panels; data presented as mean and standard error bars; LOAD: late-onset Alzheimer disease; YOAD: young-onset Alzheimer disease; TCP: tau-first cognitive proteinopathy; CTL: cognitively unimpaired controls. NFL: neurofilament light. ns: not significant. ** *p* < 0.01, *** *p* < 0.001, **** *p* < 0.0001.

**Figure 2 ijms-25-05607-f002:**
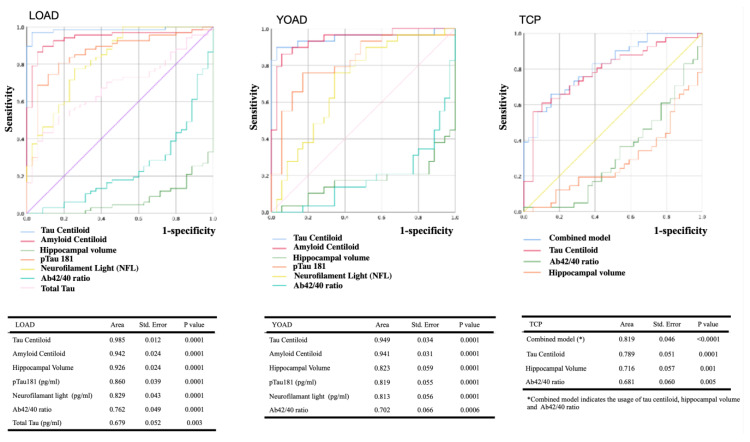
The diagnostic accuracy of the predefined biomarkers and receiver operating characteristic (ROC) curves for binary classifications (disease groups versus CTL). The tested biomarkers were tau centiloid, amyloid centiloid, hippocampal volume (average of left and right hippocampus), pTau181, total tau, NFL, and Aβ42/40 ratio. Areas under the ROC curves (AUCs) of significant biomarkers are reported. LOAD: late-onset Alzheimer disease; YOAD: young-onset Alzheimer disease; TCP: tau-first cognitive proteinopathy; CTL: cognitively unimpaired controls. NFL: neurofilament light. Std. Error: standard error.

**Figure 3 ijms-25-05607-f003:**
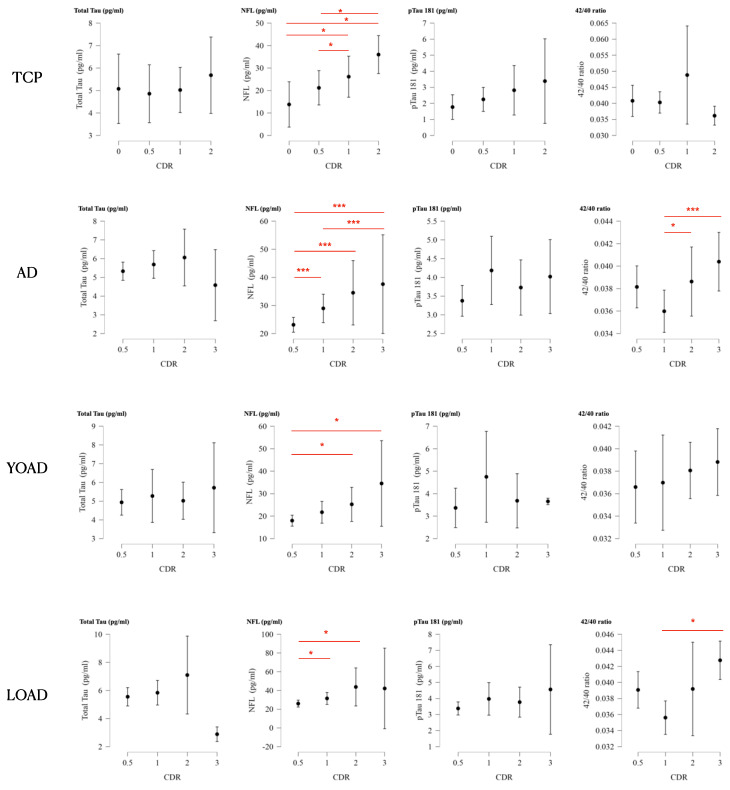
Interval plot (mean and 95% confidence interval) of clinical dementia rating (CDR) score and plasma biomarker levels. AD: Alzheimer disease; LOAD: late-onset Alzheimer disease; YOAD: young-onset Alzheimer disease; TCP: tau-first cognitive proteinopathy; NFL: neurofilament light. * *p* < 0.05, *** *p* < 0.001.

**Table 1 ijms-25-05607-t001:** Demographic data of the four study groups.

	CTL	LOAD	TCP	YOAD
Case numbers	90	96	44	55
Age at plasma, median (range)	67 (25~88)	77 (67~94) *	75.5 (51~87) *	66 (46~72) ^†^
Disease durations at plasma (year), median (range)	N.A.	4 (1~14)	3 (2~13)	4 (1~17)
Estimated year of onset, median (range)	N.A.	72 (65~85)	70.5 (49~85)	60 (43~65)
Sex, female (%)	45 (50)	62 (65)	19 (43)	34 (62)
ApoE4 carrier, *n* (%)	12, (13)	43, (45) *^†^	9, (21) *	26, (47) *^†^
Educational years	12 (0–21)	6 (0–18) *^†^	10.5 (2–16)	9 (0–16) *^†^
Mini-mental State Examination, median (range)	28 (25~30)	20 (0–28) *^†^	24 (1–28) *	21 (0–28) *^†^
CDR, median (range)	0	0.5 (0.5–3) *	0.5 (0–2) *	0.5 (0.5–3) *
Amyloid centiloid, mean (SD)	6.12 (17.2)	76.83 (47.0)*^†^	9.2 (27.31)	65.32 (36.9) *^†^
Tau centiloid, mean (SD)	2.81 (13.4)	87.24 (33.22) *^†^	28.19 (15.8) *	99.23 (50.69) *^†^
Gray matter volume (mL), mean (SD)	588.8 (60.54)	523.5 (53.03) *^†^	556.5 (55.87) *	537.0 (72.59) *^†^
Hippocampal GM volume fraction (%), mean (SD)	0.35 (0.041)	0.26 (0.048) *^†^	0.31 (0.045) *	0.28 (0.048) *

CTL: controls; TCP: tau-first cognitive proteinopathy; LOAD: late-onset Alzheimer’s disease; YOAD: young-onset Alzheimer’s disease; * *p* < 0.05 compared with CTL; ^†^ *p* < 0.05 compared with TCP.

**Table 2 ijms-25-05607-t002:** Stepwise logistic regression model for group diagnosis or differential diagnosis.

Diagnosis	Reference	Accuracy	Parameters	Estimate	z	*p*
LOAD	CTL	0.8295	Hippocampal volume	−7.6634	−5.8789	4.1311 × 10^−9^
			pTau181	0.4259	2.6842	0.0073
YOAD	CTL	0.8456	Hippocampal volume	−4.4059	−4.6756	2.9311 × 10^−6^
			pTau181	0.4110	2.5612	0.0104
			Aβ42/40	27.7192	−2.0677	0.0387
TCP	CTL	0.7236	Hippocampal volume	−3.0325	−3.4372	0.0006
**Differential Diagnosis**	**Reference**	**Accuracy**	**Parameters**	**Estimate**	**z**	** *p* **
LOAD	YOAD	0.6620	NFL	0.0393	2.5169	0.0118
TCP	LOAD	0.8062	Hippocampal volume	2.6983	3.2147	0.0013
			pTau181	−0.7275	−3.2535	0.0011
TCP	YOAD	0.7753	pTau181	−0.9556	−3.9956	6.4535 × 10^−5^

CTL: controls; TCP: tau-first cognitive proteinopathy; LOAD: late-onset Alzheimer’s disease; YOAD: young-onset Alzheimer’s disease. Stepwise logistic model with variables of plasma Aβ42/40, neurofilament light chain, pTau181, and hippocampal volume.

**Table 3 ijms-25-05607-t003:** Regression models of cognitive outcomes in each group.

Model	Parameters	Unstandardized β	*t*	*p*	95% Confident Intervals (Lower~Upper)
AD	Hippocampal volume	6.6297	4.6585	1.0831 × 10^−5^	3.8028~9.4566
	Tau centiloid	−0.0382	−3.0302	0.0032	−0.0632~−0.0132
	Educational year	0.2440	2.2417	0.0274	0.0278~0.4603
YOAD	pTau 181	−1.3113	−2.9105	0.0075	−2.2393~−0.3834
	Hippocampal volume	4.2538	2.3709	0.0258	0.5587~7.9489
LOAD	Hippocampal volume	8.3245	4.5280	2.7079 × 10^−5^	4.6506~11.9984
	Educational year	0.3338	2.4123	0.0188	0.0573~0.6104
	Tau Centiloid	−0.0417	−2.3161	0.0238	−0.0777~−0.0057
TCP	Hippocampal volume	4.9901	2.6964	0.0107	1.2330~8.7471

Dependent variables: Mini-Mental Status Examination. Independent variables: amyloid centiloid, tau centiloid, hippocampal volume, neurofilament light chain, pTau 181, Aβ42/40, total tau, and educational years. Stepwise linear regression model. AD: Alzheimer’s disease; YOAD: young-onset Alzheimer’s disease; LOAD: late-onset Alzheimer’s disease; TCP: tau-first cognitive proteinopathy.

## Data Availability

The data that support the findings of this study are available upon request from the corresponding author. The data are not publicly available due to privacy or ethical restrictions.

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
