# Peer review of "Clinical Significance of the Plasma Biomarker Panels in Amyloid-Negative and Tau PET-Positive Amnestic Patients: Comparisons with Alzheimer’s Disease and Unimpaired Cognitive Controls"

_ijms, 2024, doi:10.3390/ijms25115607_

Round 1

Reviewer 1 Report

Comments and Suggestions for Authors

Reviewer comments and suggestions

The authors in this study investigated the plasma biomarkers that could help to diagnose, differentiate from Alzheimer disease (AD), and stage cognitive performance in patients with positron emission tomography (PET)-confirmed primary age-related tauopathy, termed tau-first cognitive proteinopathy (TCP) in this study. They have enrolled the 285 subjects with young-onset AD (YOAD; n=55), late-onset AD (LOAD; n=96), TCP (n=44), and cognitively unimpaired controls (CTL; n=90), and analyzed plasma Aβ42/Aβ40, pTau181, neurofilament light (NFL) and total-tau using single molecule assays. Receiver operating characteristic curves and areas under the curves (AUCs) were used to determine the diagnostic accuracy of plasma biomarkers compared to hippocampal volume, amyloid- and tau-condyloid. For TCP, tau-centiloid reached the highest AUC for diagnosis (0.79), while pTau181 could differentiate TCP from YOAD (accuracy 0.775) and LOAD (accuracy 0.806). The study result supported the superiority of tau PET to diagnose TCP, pTau181 to differentiate TCP from YOAD or LOAD, and NFL for functional staging.

Overall, the manuscript was good. However, a few major concerns/comments needed to be explained or modified. 

  1. Line 51 NIA-AA first time used.
  2. Line 65 what does it indicate? “Primary age-related tauopathy (PART) denotes an A-T+ pathological status”
  3. Line 94 This part was not included in the introduction section previously, please explain a bit so that it could be written in the section on objectives
  4. Line 95-96 It would be nice if the authors could present a hypothesis for this study
  5. Did the authors discuss the MMSE scores
  6. I think the discussion should not be in sections as major findings or any other section. please check the guidelines of mdpi
  7. In another section I have seen the relevancy of hippocampal volume, please explain
  8. Line 243-244 Is there was possible reason for this?
  9. At various places in the discussion section, the authors may add the table or figures number in the text for reader-friendly.
  10. Almost all the references need to be modified based on the MDPI journals.

Reviewer 2 Report

Comments and Suggestions for Authors

The authors present analysis of a data-rich study to explore the utility of AD core plasma biomarkers for clinical significance in the diagnosis, staging and cognitive predictions for patients with tau-first cognitive proteinopathy. The authors established fairly solid statistical correlations between various biomarkers of neurological degenerative disease to not only accomplish the stated purpose, but to assess clinical feasibility of use of AD core plasma biomarkers to discriminate TCP from CTL, TCP from AD, and the use of hippocampal volume to predict cognitive scores.

Methods were well described. Sometimes when reviewing manuscripts, I think authors should include data from supplementary files in the main body of the published manuscript, but in this case, the authors present such a wealth of data on which to base their conclusions, that of necessity, there was a need to include a large portion in supplementary files.

The authors state limitations of the study with regard to need for a larger number of control subjects for baseline thresholds.

I have not reviewed such a data rich, well supported, well designed study in a long time.
